# The Effect of Thyroid Stimulating Hormone Level Within the Reference Range on In-Hospital and Short-Term Prognosis in Acute Coronary Syndrome Patients

**DOI:** 10.3390/medicina55050175

**Published:** 2019-05-20

**Authors:** Muhammet Gürdoğan, Servet Altay, Selçuk Korkmaz, Çağlar Kaya, Utku Zeybey, Mustafa Ebik, Melik Demir

**Affiliations:** 1Department of Cardiology, School of Medicine, Trakya University, 22030 Edirne, Turkey; svtaltay@gmail.com (S.A.); caglarkaya.2626@gmail.com (Ç.K.); utkuz1907@hotmail.com (U.Z.); mustafa.ebik@gmail.com (M.E.); melikdemir.34@gmail.com (M.D.); 2Department of Biostatistics and Medical Informatics, School of Medicine, Trakya University, 22030 Edirne, Turkey; selcukorkmaz@gmail.com

**Keywords:** acute coronary syndrome, thyroid-stimulating hormone, mortality, prognosis

## Abstract

*Background and objectives:* Despite being within the normal reference range, changes in thyroid stimulating hormone (TSH) levels have negative effects on the cardiovascular system. The majority of patients admitted to hospital with acute coronary syndrome (ACS) are euthyroid. The aim of this study was to investigate the effect of TSH level on the prognosis of in-hospital and follow-up periods of euthyroid ACS patients. *Materials and Methods:* A total of 629 patients with acute coronary syndrome without thyroid dysfunction were included in the study. TSH levels of patients were 0.3–5.33 uIU/mL. Patients were divided into three TSH tertiles: TSH level between (1) 0.3 uIU/mL and <0.90 uIU/mL (*n* = 209), (2) 0.90 uIU/mL and <1.60 uIU/mL (*n* = 210), and (3) 1.60 uIU/mL and 5.33 uIU/mL (*n* = 210). Demographic, clinical laboratory, and angiographic characteristics were compared between groups in terms of in-hospital and follow-up prognosis. *Results:* Mean age was 63.42 ± 12.5, and 73.9% were male. There was significant difference between tertiles in terms of TSH level at admission (*p* < 0.001), the severity of coronary artery disease (*p* = 0.024), in-hospital mortality (*p* < 0.001), in-hospital major hemorrhage (*p* = 0.005), total adverse clinical event (*p* = 0.03), follow-up mortality (*p* = 0.022), and total mortality (*p* < 0.001). In multivariate logistic regression analysis, the high–normal TSH tertile was found to be cumulative mortality increasing factor (OR = 6.307, 95%; CI: 1.769–22.480; *p* = 0.005) during the 6-month follow-up period after hospitalization and discharge. *Conclusions:* High–normal TSH tertile during hospital admission in euthyroid ACS patients is an independent predictor of total mortality during the 6-month follow-up period after hospitalization and discharge.

## 1. Introduction

Acute coronary syndromes are the most severe forms of cardiovascular system diseases (CVS), which are responsible for approximately one-third of worldwide deaths [1]. Although there has been a decrease in the number of deaths caused by acute coronary syndrome (ACS) in recent years thanks to many developments in diagnosis and treatment, it is still one of the leading causes of morbidity and mortality in both developed and developing countries [1,2]. Atherosclerotic plaque rupture, which causes partial or complete obstruction of an epicardial coronary artery, is the most common mechanism responsible for ACS [3]. Hypertension, hyperglycemia, and dyslipidemia are considered as major risk factors for atherosclerotic plaque formation [2]. The increased prevalence of genetically-mediated risks such as lipoprotein (a), homocysteine, fibrinogen, and plasminogen activator inhibitor, as well as factors such as intense nutrition, smoking, physical inactivity, and psychosocial stress trigger the emergence of major risk factors for the formation of atherosclerotic plaques sooner or later [1,4,5]. Although not reported among traditional risk factors for coronary artery disease (CAD), changes in circulating concentrations of thyroid hormones which have receptors in both myocardial and vascular endothelial tissues affect the cardiovascular system [6,7,8,9]. It has been reported in the literature that evident or subclinical thyroid dysfunction is associated with increased risk of vascular morbidity and mortality up to 20% to 80% [6,7,8,9]. There are a limited number of studies investigating the relationship between different thyroid stimulating hormone (TSH) levels, despite being in the normal reference range, and CVD risk and different results are observed in these studies [10,11]. When it is considered in clinical practice that the majority of patients admitted to the hospital with ACS are euthyroid, the effect of having different TSH levels on the development of adverse clinical events or mortality in these patients is not fully known [12].

In this study, it was aimed to investigate whether TSH levels of euthyroid ACS patients measured during admission were related to adverse clinical events and mortality risk during the in-hospital and post-discharge follow-up period.

## 2. Materials and Methods

### 2.1. Subject Selection 

The study was performed in the cardiology clinic of Trakya University Medical Faculty Hospital, which is the tertiary-level hospital with the highest volume of patients in the Trakya region of Turkey. This retrospective and cross-sectional study included 629 patients complying with the inclusion criteria out of a total of 831 patients with the diagnosis of ACS between January 1, 2017 and January 1, 2018. (Figure 1). 

Little’s test was performed to assess for data missing completely at random, and it was found that the missing data were missing completely at random (*p* = 0.243). The data of the patients included in the study were examined in detail, and demographic, clinical, laboratory, and angiographic parameters were recorded. Clinical event data of the patients during the in-hospital and follow-up period were obtained from patient files, the hospital digital recording system, and telephone calls with patients. The survival status of a few patients who were not reached was confirmed by the national death notification system. The study was approved by Trakya University Scientific Research Ethics Committee (TUTF-BAEK 2019/12, from 14 January 2019).

### 2.2. Collection of Data

Acute coronary syndrome was defined as the presence of unstable angina pectoris, non-ST segment-elevation, or ST-elevation myocardial infarction [13]. Body mass index (BMI) was calculated as body weight (kg)/square of the height (m^2^) [14]. Glomerular filtration rate (GFR) was calculated using the Cockcroft–Gault formula; [(140 − age) × body weight (kg)]/[72 × serum creatinine] (if women ×0.85) [15]. CAD was diagnosed by a history of myocardial infarction and revascularization, by angiographic greater than 50% stenosis of at least one major coronary artery [16]. Transfusion of at least 2 units of blood, reduction in the hemoglobin level of at least 2 g/L, or symptomatic bleeding in a critical area was determined as major bleeding [17]. While in-hospital events of the study were in-hospital mortality, major bleeding, re-infarct, life-threatening arrhythmias, cardiopulmonary arrest, acute renal failure, acute heart failure, mechanic complications inotropic drug requirement, and stroke, the follow-up period total adverse events were follow-up mortality, re-hospitalization, re-infarct, and need for target vessel revascularization. Thyroid hormone levels of the patients included in the study were studied with a Unicel DXI 600 (Beckman Coulter, Porterville, CA, USA) device by a chemical immunoassay method. Patients with hyperthyroidism, hypothyroidism, and euthyroidism were determined by considering the limit values for thyroid hormone levels (TSH: 0.3–5.33 uIU/mL, fT3: 1.4–5 pg/mL, fT4: 0.6–1.62 ng/dL). The patients with low TSH levels and high free T3 and T4 levels were defined as evident hyperthyroidism, and the patients with low TSH levels and normal free T3 and T4 levels were defined as subclinical hyperthyroidism. The patients with high TSH levels and low free T3 and T4 levels were defined as evident hypothyroidism, and the patients with high TSH levels and normal free T3 and T4 levels were defined as subclinical hypothyroidism [18]. A total of 629 euthyroid patients with a diagnosis of ACS and with a normal reference range of TSH were grouped in accordance with their TSH levels and were divided into three tertiles to have an equal number of patients, similar to other studies in the literature [19,20,21,22]. Based on TSH tertile grouping, (1) 209 patients with TSH level of 0.3 uIU/m and <0.90 uIU/mL, (2) 210 patients with TSH level of 0.90 uIU/mL and <1.60 uIU/mL, and (3) 210 patients with TSH level of 1.60 uIU/mL’den 5.33 uIU/mL were included.

### 2.3. Statistical Analysis 

Taking the study by Wanjia et al. as a reference, the power analysis was performed, and the effect size was calculated as 0.125 [20]. It was decided to include at least 621 patients at the calculated effect size, 80% power, and 5% significance level. The normal distribution hypothesis was tested via the Shapiro–Wilk test. Group comparisons were performed through one-way analysis of variance for variables with normal distribution, while group comparisons for variables without normal distribution were performed via the Kruskal-Wallis test. The Tukey test and Siegel–Castellan test were used for multiple comparisons. The associations between qualitative variables were investigated by Pearson χ-square and Fisher’s exact test. Ordered logistic regression was used to adjust the gender for analyses regarding TSH tertiles. Multivariate logistic regression analysis was used to investigate the risk factors affecting mortality. Demographic variables included age, gender, smoking, clinical variables including hyperlipidemia and hypertension, coronary artery disease, laboratory and angiographic parameters, including hemoglobin, albumin, aspartate aminotransferase, glomerular filtration rate, left ventricular ejection fraction, multivessel disease, and adverse event development in-hospital variable, including major bleeding, and the multivariate logistic regression model was adjusted for these variables. Descriptive statistics were given as mean ± standard deviation and the median and quartiles for the quantitative variables, while the percent and frequency were given as descriptive statistics for the qualitative variables. SPSS 20.0 (IBM SPSS Statistics for Windows, version 20.0; Armonk, NY: IBM Corp., USA) was used for the analysis of the data. The significance level was determined as 0.05 in all statistical analyzes.

## 3. Results 

The mean age of the 629 patients included in the study was 63.42 ± 12.5, and 73.9% were male. The mean length of hospital stay in the coronary intensive care unit was 4.07 ± 4.61, and the mean total hospitalization period was 7.86 ± 7.3 days. There was no significant difference between the tertiles in basal demographic and clinical characteristics of the patients except for gender (Table 1). When the laboratory and angiographic data of the patients during admission were examined, there was significant difference between tertiles in the albumin (*p* = 0.036), hemoglobin (*p* = 0.048), aspartate transaminase (*p* = 0.026), left ventricle ejection fraction (LVEF) (*p* = 0.02), TSH levels (*p* < 0.001), and the severity of coronary artery disease (*p* = 0.024), while there was no significant difference between tertiles in the other parameters (Table 2). There was no difference between tertiles in terms of the treatments given to patients during the in-hospital period (Table 3). During the in-hospital period, mortality (*p* < 0.001), major hemorrhage (*p* = 0.005), and total adverse clinical event (*p* = 0.03) were significantly higher in the high–normal TSH tertile, while no significant difference was observed in terms of other adverse events. In the follow-up period, mortality (*p* = 0.022) and total adverse clinical event (*p* = 0.006) and cumulative mortality during the in-hospital and follow-up period (*p* < 0.001) were found to be significantly higher in the high–normal TSH tertile (Table 4). The factors that differed between the groups and the variables that were accepted as traditional risk factors for CAD were included in multivariate logistic regression, and high–normal TSH level did not have a statistically significant effect on in-hospital mortality (odds ratio (OR) = 5.218, 95% confidence interval (CI): 0.773–35.211, *p* = 0.09). However, LVEF (OR = 0.942, 95%; CI: 0.890–0.998, *p* = 0.043) and albumin (OR = 0.267, 95% CI: 0.070–1.014, *p* = 0.052) had protective effects in in-hospital mortality (Table 5). The predictors for follow-up mortality were only high TSH level (OR = 5.867, 95% CI: 1.083–31.774, *p* = 0.04) and diabetes (OR = 4.148, 95% CI: 1.243–13.841, *p* = 0.021). In multivariate logistic regression analysis performed to determine cumulative mortality determinants, high TSH level was found to be a mortality risk-increasing factor (OR = 6.307, 95% CI: 1.769–22.480, *p* = 0.005). Furthermore, LVEF (OR = 0.956, 95% CI: 0.916–0.999, *p* = 0.043) and albumin (OR = 0.252, 95% CI: 0.099–0.642, *p* = 0.004) were found to have protective effects, similar to in-hospital mortality. (Table 5)

## 4. Discussion 

The most important findings of the study can be listed as follows: (1) High–normal TSH level in patients with euthyroid ACS is an independent predictor of total mortality during the 6-month follow-up period after hospitalization and discharge; (2) high–normal TSH level in euthyroid ACS patients is associated with the severity of coronary artery disease; and (3) in euthyroid ACS patients, there is an association between the high–normal TSH tertile and total adverse clinical event incidence, especially including major hemorrhage in-hospital and during the follow-up period.

In the literature, there are different results regarding the effect of having a low–normal or high–normal TSH tertile on cardiovascular mortality. In the HUNT study (Nord-Trøndelag Health Study) with a 12-year follow-up period by Asvold et al., it was reported that despite being in the reference range, women with high serum TSH levels have an increased risk of CAD mortality [11]. In another study of 14 cohorts by the same investigators, TSH levels in the reference range were not associated with increased risk of coronary heart disease mortality [11,23]. In another study by Altay et al., it was reported that high–normal TSH level was associated with all-cause mortality and morbidity in female patients only, and this result was attributed to pro-inflammatory status [19]. In a 10-year cohort study by Parle et al., it was reported that although it is within the normal reference range in healthy subjects aged 60 years and older, the low–normal TSH tertile is associated with an increase in both all-cause and cardiovascular-induced mortality [10]. In a study conducted by Onat et al., it was found that low–normal serum TSH tertile in middle-aged euthyroid adults in the general population was associated with coronary heart disease, especially in males, regardless of traditional cardiovascular risk factors or presence of metabolic syndrome [12]. In another study conducted by Li et al., it was reported that there was a 4-fold higher risk of coronary heart disease in individuals with high–normal TSH tertile compared to those with low–normal TSH tertile [24].

As it is seen, in community-based studies, there are controversial results on the association between low–, moderate– or high–normal TSH tertile state and mortality risk due to coronary artery disease. In our studyl however, when compared with the low–normal TSH level in euthyroid ACS patients, it was seen that the moderate–normal TSH tertile increased mortality by 2.9 times and the high–normal TSH tertile by 6.3-fold. This situation can be interpreted as a linear relationship between TSH tertiles and mortality in ACS patients with TSH level in the normal reference range. The significance of our study is that it differs from the literature in terms of investigating the relationship of TSH levels in a normal reference range with mortality in ACS patients. The results of our study indicate that CAD is more common in high–normal TSH tertile as well as the increase in mortality. Similar to our study, Ndrepepa et al. reported that in euthyroid patients with stable CAD who underwent percutaneous coronary intervention, CAD was more prevalent in the group with high–normal TSH tertile [20]. In this study, it was also found that high–normal TSH levels were associated with a 3-year increase in all-cause mortality. Our study shows that high levels of normal TSH are associated with both the severity of CAD and short-term mortality in patients with ACS.

The high–normal TSH level has been reported to be one step ahead of an early stage of hypothyroidism development [25]. Studies have shown that hypothyroidism, whether subclinical or evident, has many adverse effects on CVS [6,7,8,9,26,27,28,29,30,31]. It is reported that evident hypothyroidism with a prevalence of 4% to 10% in the community is closely associated with hypercholesterolemia, diastolic hypertension, increased carotid intima-media thickness, systemic inflammation, hyperhomocysteinemia, impaired fibrinolysis and platelet abnormalities, hypercoagulability, and reduction in the production of nitric oxide, a relaxation factor of endothelial origin, all of which are well-known risk factors for atherosclerosis [6,7,8,32]. The prevalence of subclinical hypothyroidism is between 4% and 20% in the community [30]. Subclinical hypothyroidism has been reported to be associated with diastolic hypertension, increased total and low-density lipoprotein (LDL) cholesterol levels, impaired endothelium-originated vasodilatation, increased arterial stiffness, and impaired left ventricular systolic and diastolic functions [8,9,29,30,31]. Therefore, the increase in cardiovascular mortality in the high–normal TSH tertile found in our study may be due to cardiometabolic disorders occurring in subclinical or evident hypothyroidism. It is a known fact that inflammation plays an important role in the pathogenesis and prognosis of CAD and ACS [33,34,35]. TSH is an important stimulant for pro-inflammatory cytokine release in adipose tissue [36]. In parallel, subclinical hypothyroidism has been shown to be associated with chronic low-grade inflammation and endothelial dysfunction [37]. The results of our study along with the findings in the literature suggest that the high–normal TSH level may be related to severe CAD and mortality by means of chronic low-grade inflammation.

Another remarkable point in our study was that the incidence of in-hospital major hemorrhage in euthyroid ACS patients with high–normal TSH was significantly higher than that of patients with low–normal and moderate–normal TSH. A study by Yango et al. reported that the change in the synthesis and activity of clotting factors such as von Willebrand factor and factor VIII in case of hypothyroidism increased the tendency for hemorrhage [38]. Therefore, this may be the reason for the high frequency of hemorrhage in the patient group with high–normal TSH tertile.

### Limitations 

The results of this retrospective study using single center data should be interpreted with some limitations. Although TSH measurements used in the study were taken within the first 24–48 h period after admission, this varies from patient to patient. In addition, thyroid hormone levels were determined based on a single measurement value and were not repeated. Nutritional status of patients which affects thyroid hormone levels could not have been queried in our study. Moreover, the short follow-up period and the lack of evaluation of possible differences based on age and gender can be considered as other limitations. Although it has been shown in the study that being in different TSH tertiles within the reference range is associated with an increase in cardiovascular mortality, it cannot be determined clearly which mechanisms this association is related to.

## 5. Conclusions

This study shows that high–normal TSH levels in euthyroid ACS patients are associated with an increase in the frequency of adverse clinical events and mortality during the 6-month follow-up period after hospitalization and discharge. Despite being in the normal reference range, the high–normal TSH tertile can be a predictor for guiding a closer control of cardiovascular risk factors to prevent the development of adverse clinical events, including death in ACS patients. This correlation will be clarified by prospective studies with large patient numbers and longer follow-up.

## Figures and Tables

**Figure 1 medicina-55-00175-f001:**
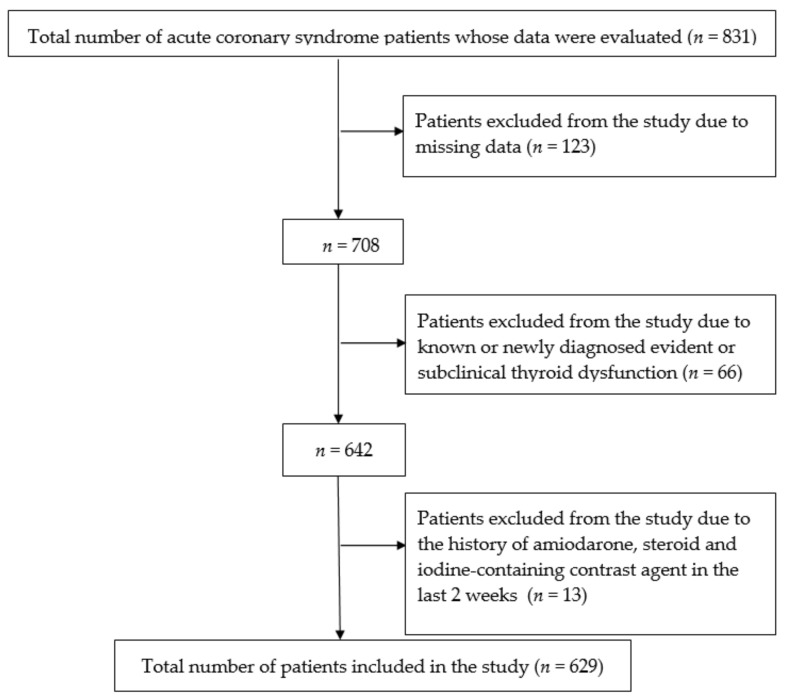
Inclusion and exclusion criteria.

**Table 1 medicina-55-00175-t001:** Comparison of groups in terms of baseline demographic and clinical features.

Variables		Low Tertile (*n* = 209)	Mid Tertile (*n* = 211)	High Tertile (*n* = 209)	*p* Value
Age (year)		63.00 ± 11.26	62.53 ± 13.39	64.76 ± 12.90	0.179
Gender	Male	163 (35.10)	164 (35.30)	138 (29.70)	0.006 *
	Female	46 (28.00)	47 (28.70)	71 (43.80)	
BMI, kg/cm^2^		27.40 ± 2.83	27.40 ± 2.90	27.63 ± 3.79	0.874
Smoking		113 (33.70)	115 (34.30)	107 (31.90)	0.562
Hypertension		113 (30.80)	125 (34.10)	129 (35.10)	0.269
Diabetes mellitus		77 (37.70)	65 (31.90)	62 (30.40)	0.242
Hyperlipidemia		21 (28.40)	30 (40.50)	23 (31.10)	0.380
Heart failure		10 (31.20)	9 (28.10)	13 (40.60)	0.641
Cancer		22 (40.00)	16 (29.10)	17 (30.90)	0.526
CAD		74 (34.10)	74 (34.10)	69 (31.80)	0.856
CVA		18 (40.90)	12 (27.30)	14 (31.80)	0.491
PAD		3 (37.50)	2 (25.00)	3 (37.50)	0.827
CKD		16 (37.20)	11 (25.60)	16 (37.20)	0.519
COPD		14 (30.40)	21 (45.70)	11 (23.90)	0.167
AF		11 (31.40)	13 (37.10)	11 (31.40)	0.898
SBP, mmHg		121.35 ± 18.00	119.32 ± 15.48	121.07 ± 17.54	0.458
DBP, mmHg		72.98 ± 13.02	72.60 ± 11.42	74.05 ± 10.47	0.458
ACS subtypes	UA or NSTEMI	125 (33.70)	121 (32.60)	125 (33.70)	0.839
	STEMI	84 (32.60)	90 (34.90)	84 (32.60)	

* Statistical significance (*p* < 0.05); descriptives: Mean ± standard deviation (SD), count (percent). Abbreviations: BMI: Body mass index, CAD: Coronary artery disease, CVA: Cerebrovascular accident, PAD: Peripheral arterial disease, CKD: Chronic kidney disease, COPD: Chronic obstructive pulmonary disease, AF: Atrial fibrillation, SBP: Systolic blood pressure, DBP: Diastolic blood pressure, ACS: Acute coronary syndrome, UA: Unstable angina, NSTEMI: Non-ST segment elevation myocardial infarction, STEMI: ST segment elevation myocardial infarction.

**Table 2 medicina-55-00175-t002:** Comparison of groups according to laboratory and angiographic parameters at the admission.

Laboratory Analysis	Low Tertil (*n* = 209)	Mid Tertil (*n* = 211)	High Tertil (*n* = 209)	*p* Value
WBC count, 10^9^/L	11.10 (8.60–14.10)	10.00 (8.40–13.17)	10.20 (8.30–13.00)	0.095
Lymphocyte, 10^9^/L	2.00 (1.40–2.70)	2.10 (1.50–2.70)	2.00 (1.30–3.00)	0.864
Hemoglobin, g/dL	13.44 ± 2.32	13.58 ± 2.06	13.07 ± 2.22	0.048 *
CRP	0.70 (0.30–2.02)	0.54 (0.30–1.30)	0.70 (0.30–2.74)	0.073
Glucose, mg/dL	129 (108–192)	129 (103–172)	128 (108.5–172)	0.653
GFR, mL/min	93 (74.75–107.25)	88.5 (65.75–111)	81.5 (64–106.25)	0.054
TG, mg/dL	122 (86.25–171.75)	131 (90.50–205)	131 (100–176.75)	0.340
TC, mg/dL	184.14 ± 49.23	184.00 ± 48.09	183.83 ± 47.67	0.998
LDL-C, mg/dL	119.61 ± 37.31	117.38 ± 34.86	119.74 ± 35.51	0.755
Total bilirubin, mg/dL	0.50 (0.30–0.70)	0.50 (0.30–0.70)	0.50 (0.30–0.875)	0.835
ALT, IU/L	21 (17–32)	21 (14–28.5)	22 (14–31)	0.074
AST, IU/L	36 (25–73)	30 (23–50)	34 (24–57)	0.026 *
INR	1.13 ± 0.27	1.12 ± 0.31	1.16 ± 0.28	0.290
APTT	29 (26.625–34)	29.05 (26.92–32.57)	30 (27.05–34)	0.629
Albumin	3.68 ± 0.54	3.75 ± 0.52	3.60 ± 0.52	0.036 *
Sodium	137.43 ± 3.45	137.68 ± 3.17	137.41 ± 3.60	0.662
Potassium	4.46 ± 0.56	4.39 ± 0.52	4.36 ± 0.64	0.195
Calcium	8.97 ± 0.64	9.04 ± 0.77	8.90 ± 0.84	0.475
Magnesium	1.93 ± 0.40	1.94 ± 0.31	1.97 ± 0.34	0.860
FT3	2.97 ± 0.49	2.95 ± 0.61	2.97 ± 0.57	0.949
FT4	0.90 (0.8–1.0825)	0.90 (0.8–1.05)	0.90 (0.8–1.09)	0.896
TSH	0.60 ± 0.19	1.25 ± 0.20	2.61 ± 0.93	<0.001*
LVEF, %	50.86 ± 9.00	52.52 ± 9.33	49.81 ± 10.65	0.020 *
Extend of CAD *				
One-vessel disese	76 (39.2)	66 (34)	52 (26.8)	0.024 *
Multivessel disease	109 (29.1)	129 (34.5)	136 (36.4)	

* Statistical significance (*p* < 0.05); descriptives: Mean ± SD, median (25th–75th percentile), count (percent). Abbreviations: WBC: White blood cell, CRP: C-reactive protein, GFR: Glomerular filtration rate, TG: Triglycerides, TC: Total cholesterol, LDL-C: Low-density lipoprotein–cholesterol, ALT: Alanine aminotransferase, AST: Aspartate aminotransferase, INR: International normalized ratio, APTT: Activated partial thromboplastin time, FT3: Free T3 (triiodothyronine), FT4; Free T4 (thyroxine), TSH: thyroid stimulating hormone, LVEF: Left ventricular ejection fraction, CAD: Coronary artery disease. *The number of coronary angiography applied subjects (*n* = 568).

**Table 3 medicina-55-00175-t003:** Comparison of groups according to the drug treatment given in the hospital.

In-Hospital Medication	Low Tertil (*n* = 209)	Mid Tertile (*n* = 211)	High Tertil (*n* = 209)	*p* Value
ASA	200 (32.60)	208 (33.90)	205 (33.40)	0.134
Clopidogrel	161 (33.40)	161 (33.40)	160 (33.20)	0.984
Ticagrelor	33 (29.70)	39 (35.10)	39 (35.10)	0.689
Prasugrel	5 (41.70)	5 (41.70)	2 (16.70)	0.511
Beta-blockers	151 (31.90)	167 (35.20)	156 (32.90)	0.249
CCB (dihydropyridine)	11 (27.50)	17 (42.50)	12 (30.00)	0.455
ACEI	125 (32.90)	134 (35.30)	121 (31.80)	0.519
ARB	3 (50.00)	3 (50.00)	0 (0.00)	0.256
Heparin (all class)	194 (33.80)	193 (33.60)	187 (32.60)	0.475
Statins	180 (32.80)	189 (34.40)	180 (32.80)	0.472
Furosemid	22 (39.30)	13 (23.20)	21 (37.50)	0.226
Spironolactone	9 (27.30)	11 (33.30)	13 (39.40)	0.680
Nitrates	45 (36.00)	37 (29.60)	43 (34.40)	0.563
Inotropic agents	14 (26.40)	17 (32.10)	14 (26.40)	0.361

Descriptives: Count (percent). Abbreviations: ASA: Acetylsalicylic acid, CCB: Calcium channel blocker, ACEI: Angiotensin converting enzyme inhibitor, ARB: Angiotensin receptor blocker.

**Table 4 medicina-55-00175-t004:** Comparison of groups in terms of adverse event development in-hospital and follow-up period.

In-Hospital Period	Low Tertile (*n* = 209)	Mid Tertile (*n* = 211)	High Tertile (*n* = 209)	*p* Value
CICU LOS, days	3 (2–4)	3 (2–4)	3 (2–5)	0.221
Total LOS, days	6 (4–9)	6 (5–8)	6 (4–9)	0.844
In-hospital event	52 (32.30)	43 (26.70)	66 (41.00)	0.03 *
Major bleeding	5 (20.00)	4 (16.00)	16 (64.00)	0.005 *
In-hospital transfusion	8 (21.10)	14 (36.80)	16 (42.10)	0.235
Re-infarct	1 (50.00)	0 (0.00)	1 (50.00)	0.553
VT/VF	7 (41.20)	2 (11.80)	8 (47.10)	0.149
Heart failure	6 (40.00)	3 (20.00)	6 (40.00)	0.531
New onset AF	8 (34.80)	6 (26.10)	9 (39.10)	0.718
Acute renal failure	8 (23.30)	6 (30.00)	9 (46.70)	0.252
CPA	17 (35.40)	13 (27.10)	18 (37.50)	0.613
Mechanic complications	4 (44.40)	3 (33.30)	2 (22.20)	0.783
Infection	15 (27.30)	14 (25.50)	26 (47.30)	0.067
In-hospital mortality	5 (12.20)	12 (29.30)	24 (58.50)	<0.001 *
Inotropic drug requirement	12 (25.00)	15 (31.20)	21 (43.80)	0.238
Stroke	1 (50.00)	1 (50.00)	0 (0.00)	1.000
**Follow-up period (6 months)**				
Total event	37 (34.60)	23 (21.50)	47 (43.90)	0.006 *
Mortality	13 (28.90)	9 (20.00)	23 (51.10)	0.022 *
Re-hospitalization	12 (28.60)	10 (23.80)	20 (47.60)	0.112
Re-infarct	22 (33.30)	18 (27.30)	26 (39.40)	0.425
Cumulative mortality	18 (20.90)	21 (24.40)	47 (54.70)	<0.001 *

* Statistical significance (*p* < 0.05); Descriptives: Median (25th–75th percentile), count (percent). Abbreviations: CICU: Coronary intensive care unit, LOS: Length of stay, VT: Ventricular tachycardia, VF: Ventricular fibrillation, AF: Atrial fibrillation, CPA: Cardiopulmonary arrest.

**Table 5 medicina-55-00175-t005:** Multivariate logistic regression analysis of in-hospital, follow-up, and cumulative mortality for patients with acute coronary syndrome.

Variable	In Hospital Mortality		Follow-Up Mortality		Cumulative Mortality	
	OR (95% CI)	*p* Value	OR (95% CI)	*p* Value	OR (95% CI)	*p* Value
TSH (mid)	2.542	0.356	3.614	0.146	2.965	0.101
	(0.351–18.429)		(0.639–20.428)		(0.808–10.883)	
TSH (high)	5.218	0.09	5.867	0.040 *	6.307	0.005 *
	(0.773–35.211)		(1.083–31.774)		(1.769–22.480)	
Age	1.001	0.969	0.989	0.669	0.993	0.735
	(0.946–1.059)		(0.941–1.040)		(0.954–1.033)	
Gender (Male)	1.178	0.837	1.104	0.88	1.18	0.762
	(0.249–5.581)		(0.305–4.000)		(0.405–3.434)	
DM	0.906	0.883	4.148	0.021 *	2.398	0.069
	(0.245–3.357)		(1.243–13.841)		(0.934–6.156)	
Smoking	0.662	0.528	1.044	0.943	0.761	0.559
	(0.184–2.383)		(0.324–3.366)		(0.304–1.904)	
CAD	2.451	0.171	0.587	0.391	1.126	0.803
	(0.680–8.839)		(0.174–1.982)		(0.444–2.855)	
HL	0.885	0.901	1.136	0.873	1.127	0.859
	(0.128–6.121)		(0.239–5.386)		(0.302–4.212)	
LVEF	0.942	0.043 *	0.976	0.409	0.956	0.043 *
	(0.890–0.998)		(0.922–1.034)		(0.916–0.999)	
Multivessel	0.624	0.499	0.379	0.097	0.424	0.072
	(0.159–2.446)		(0.121–1.192)		(0.167–1.080)	
HGB	1.095	0.546	0.93	0.578	0.998	0.983
	(0.815–1.471)		(0.719–1.202)		(0.815–1.222)	
GFR	0.992	0.445	1	0.919	0.997	0.563
	(0.973–1.012)		(0.991–1.008)		(0.987–1.007)	
AST	1.003	0.397	0.998	0.657	1.001	0.705
	(0.973–1.012)		(0.989–1.007)		(0.996–1.007)	
Albumin	0.267	0.052	0.35	0.065	0.252	0.004 *
	(0.070–1.014)		(0.115–1.069)		(0.099–0.642)	
Major Bleeding	0.448	0.523	0.378	0.426	0.333	0.25
	(0.038–5.267)		(0.035–4.140)		(0.051–2.166)	
HT	0.739	0.681	1.246	0.754	0.943	0.912
	(0.175–3.120)		(0.315–4.929)		(0.330–2.692)	

* Statistical significance (*p* < 0.05); OR: Odds ratio, CI: Confidence interval, Abbreviations: TSH: Thyroid stimulating hormone, DM: Diabetes mellitus, CAD: Coronary artery disease, HL: Hyperlipidemia, LVEF: Left ventricular ejection fraction, HGB: Hemoglobin, GFR: Glomerular filtration rate, AST: Aspartate aminotransferase, HT: Hypertension.

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
