# Peer review of "The Effect of Thyroid Stimulating Hormone Level Within the Reference Range on In-Hospital and Short-Term Prognosis in Acute Coronary Syndrome Patients"

_medicina, 2019, doi:10.3390/medicina55050175_

Reviewer 1 Report

Gurdogan et al wrote an interesting retrospective study on the association between TSH levels and short- and long-term outcome in patients with Acute Coronary Syndrome.

Comments:

The authors have chosen to subdivide TSH levels into three groups with the third group consisting of TSH between 1.60 and 5.33 uIU/ml. I would be interested to know if they can come up with an cut-off value of a TSH values that can be associated with an unfavourable outcome, as the range of TSH values in this group is still wide.

The authors have found an association between TSH levels and outcome in Acute Cornary Syndrome but they do not come up with implications for clinical care of these patients. Should TSH mesurement be part of the standard care of patients with ACS? And what should be the consequence of a normal TSH between 1.6 en 5.33 uIU/ml? The authors should discuss this in their discussion.

Author Response

Response to Reviewer 1 Comments

Many thanks for your didactic and comprehensive evaluation of the manuscript. Below, you will find our responses to the Reviewer-1:

Point 1: The authors have chosen to subdivide TSH levels into three groups with the third group consisting of TSH between 1.60 and 5.33 uIU/ml. I would be interested to know if they can come up with an cut-off value of a TSH values that can be associated with an unfavourable outcome, as the range of TSH values in this group is still wide. The authors have found an association between TSH levels and outcome in Acute Cornary Syndrome but they do not come up with implications for clinical care of these patients. Should TSH mesurement be part of the standard care of patients with ACS? And what should be the consequence of a normal TSH between 1.6 en 5.33 uIU/ml? The authors should discuss this in their discussion.

Response 1: 

Only patients with normal TSH levels were included in this study. The aim of this study was not to find a cut-off value for TSH. Therefore our dataset was not suitable to find a cut-off value for TSH values. Similar studies in the literature did not an cut-off value (please see ref.12,19,24)

Our study results suggest “high-normal TSH tertile during hospital admission in ACS patients is an independent predictor of total mortality”. We have indicated this sentence in conclusion section. “Despite being in the normal reference range, the high-normal TSH tertile can be a predictor for guiding a closer control of cardiovascular risk factors to prevent the development of adverse clinical events, including death in ACS patients” (Lines: 290-293)

Reviewer 2 Report

General comments:

In this study, the authors analyzed a cohort of 629 euthyroid patients that had ACS. The designated outcomes of interest were various clinical variables and mortality endpoints measured at different timeframes. Authors hypothesized that higher levels of TSH within the euthyroid spectrum could lead to more adverse clinical outcomes and mortality. Analysis per TSH groups stratified by tertiles showed that there were indeed differences between these groups in terms of CAD severity, in-hospital mortality, in-hospital hemorrhage, total clinical events, follow-up mortality, and total mortality. Multivariable logistic regression showed that high TSH levels within the euthyroid spectrum were associated with an increased odds of cumulative mortality during the 6-month post-discharge. Authors concluded that higher TSH levels among euthyroid ACS patients independently predict cumulative mortality after 6-month FU.

Ethical/legal disclosures:

Ethical approval for this study has been obtained and is declared in the manuscript.

Statistical methodology:

Statistical methods seem appropriate. Authors provided a flow-chart showing patient exclusion and inclusion criteria.

Specific comments:

-          Authors state that 831 patients with ACS were evaluated while even 123 patients were excluded prior to enrollment in the analysis. That is about 15% of the total sample. Authors should provide evidence that these excluded patients did not systematically differ from those that were included in terms of baseline comorbidities, laboratory variables, and clinical profiles. Authors should provide an explanation of whether data that was missing is missing at random.

-          It is known that women tend to have thyroid dysfunction more frequently than men, and as we can see in Table 1., high tertile group had the highest proportion of women, and this was significant. Authors should define that all subsequent analyses regarding TSH were sex-adjusted.

-          Authors should state which variables were included in the multivariable regression analysis. It is not clear from the body of the manuscript (Methods) and is also not explained in the Table legend.

-          Table 1., Table 3 and Table 4. Please remove row NO from all tables, and leave only YES rows. We want to know the percentage and comparisons of positive traits, not the frequency of absent traits. This is not how this data should be presented.

-          Table 4, and throughout the text. Authors should define what does in-hospital event mean, what does total adverse events mean, etc. All of these should be precisely explained in the Methods. How was major bleeding defined in your study?

-          Table 4. Replace „inotropic“ with „inotropic“

-          Table 5. It should be stated in the Methods and Statistical section, what variables were multivariable model adjusted for.

-          Table 5. Remove columns „Coefficient (SE)“. It is redundant.

-          Table 6. Remove „Coefficient (SE)“ column

-          The discussion is well-rounded and touches important points

-          Relevant limitations are listed by the authors

Author Response

Response to Reviewer 2 Comments

Many thanks for your didactic and comprehensive evaluation of the manuscript. Below, you will find our responses to the Reviewer-2:

General comments:

In this study, the authors analyzed a cohort of 629 euthyroid patients that had ACS. The designated outcomes of interest were various clinical variables and mortality endpoints measured at different timeframes. Authors hypothesized that higher levels of TSH within the euthyroid spectrum could lead to more adverse clinical outcomes and mortality. Analysis per TSH groups stratified by tertiles showed that there were indeed differences between these groups in terms of CAD severity, in-hospital mortality, in-hospital hemorrhage, total clinical events, follow-up mortality, and total mortality. Multivariable logistic regression showed that high TSH levels within the euthyroid spectrum were associated with an increased odds of cumulative mortality during the 6-month post-discharge. Authors concluded that higher TSH levels among euthyroid ACS patients independently predict cumulative mortality after 6-month FU.

Ethical/legal disclosures:

Ethical approval for this study has been obtained and is declared in the manuscript.

Statistical methodology:

Statistical methods seem appropriate. Authors provided a flow-chart showing patient exclusion and inclusion criteria.

Specific comments:

Point 1: Authors state that 831 patients with ACS were evaluated while even 123 patients were excluded prior to enrollment in the analysis. That is about 15% of the total sample. Authors should provide evidence that these excluded patients did not systematically differ from those that were included in terms of baseline comorbidities, laboratory variables, and clinical profiles. Authors should provide an explanation of whether data that was missing is missing at random.

Response 1:

We added the following sentences to subject selection “There were no statistical differences in terms of basal comorbidities, laboratory variables and clinical profiles in included and excluded patients. Little’s test is performed to assess for missing completely at random and it is found that the missing data are missing completely at random (p=0.243).” (Lines: 71-75)

The comparative tables of the excluded and included patients are included at the end of this response for your information (please see appendix).

Point 2: It is known that women tend to have thyroid dysfunction more frequently than men, and as we can see in Table 1., high tertile group had the highest proportion of women, and this was significant. Authors should define that all subsequent analyses regarding TSH were sex-adjusted.

Response 2:

Ordered logistic regression is used to adjust the sex for analyses in Table 1-2-3-4.

The following sentence is added to the lines between 123-124 in the Statistical Analysis section: “Ordered logistic regression is used to adjust the gender for analyses regarding TSH tertiles.”

Point 3: Table 1., Table 3 and Table 4. Please remove row NO from all tables, and leave only YES rows. We want to know the percentage and comparisons of positive traits, not the frequency of absent traits. This is not how this data should be presented.

Response 3:

We have revised Table 1,2 and 3 based on the reviewer’s suggestion.

Point 4: Table 4, and throughout the text. Authors should define what does in-hospital event mean, what does total adverse events mean, etc. All of these should be precisely explained in the Methods.

In-hospital event and total adverse events are explained in the section on methods. (Lines: 92-97)

How was major bleeding defined in your study?

Major bleeding was defined as follows. Added to section Collection of Data. (Lines: 91-92)

References section was revised. (Lines: 353-356)

"Transfusion of at least 2 units of blood, reduction in the hemoglobin level of at least 2 g/L, or symptomatic bleeding in a critical area were determined as major bleeding."

Point 5: Table 4. Replace „inotropic“ with „inotropic“

Response 5:

Table 4 was revised according to your suggestions.

Point 6 :

Authors should state which variables were included in the multivariable regression analysis. It is not clear from the body of the manuscript (Methods) and is also not explained in the Table legend.

Table 5. It should be stated in the Methods and Statistical section, what variables were multivariable model adjusted for.

Response 6:

Table 5 and Table 6 are merged in order to avoid complexity and confusion.

The following sentence is added to the lines between 125-130 in the Statistical Analysis section.

"Demographic variables including age, gender, smoking, clinical variables including hyperlipidemia and hypertension, coronary artery disease, laboratory and angiographic parameters including hemoglobin, albumin, aspartate aminotransferase, glomerular filtration rate, left ventricular ejection fraction, multivessel disease and adverse event development in-hospital variable including major bleeding, and the multivariate logistic regression model is adjusted for these variables."

Point 7: Table 5. Remove columns „Coefficient (SE)“. It is redundant.

Response 7:

Table 5 was revised according to recommendations and combined with Table 6.

Point 8:

-Table 6. Remove „Coefficient (SE)“ column

Response 8:

We removed Coefficient (SE) column in Table 6 and we combined Table 5 and Table 6, as Table 5. 

-The discussion is well-rounded and touches important points

-Relevant limitations are listed by the authors

Appendix.

Table 1. Comparison of patients included and excluded in terms of baseline demographic and clinical features.

Variables

Patients included (n=629)

Patients excluded (n=123)

p   value

Age   (year)

63.43±12.57

62.17±10.81

0.300

Gender (Male)

465   (73.90)

97   (78.90)

0.249

BMI,   kg/cm2

27.48±3.22

27.58±2.23

0.849

Smoking

335 (53.30)

67 (54.50)

0.968

Hypertension

367   (58.30.80)

71   (57.70)

0.898

Diabetes mellitus

204   (32.40)

39   (31.70)

0.875

Hyperlipidemia

74   (11.80)

15   (12.20)

0.892

Heart failure

32   (5.10)

6   (4.90)

0.923

Cancer

55   (8.70)

8   (6.50)

0.412

CAD

217   (34.50)

44   (35.77)

0.786

CVA

44   (7.00)

6   (4.90)

0.389

PAD

8   (1.30)

5   (4.10)

0.073

CKD

43   (6.80)

5   (4.10)

0.250

COPD

46   (7.30)

10   (8.10)

0.752

AF

35   (5.60)

9   (7.30)

0.449

SBP,   mmHg

120.56±17.02

121.98±17.70

0.437

DBP,   mmHg

73.21±11.65

72.97±10.78

0.350

ACS subtypes (STEMI)

84   (41.00)

90   (40.70)

0.940

Descriptives: mean ±SD, count (percent). Abbreviations: BMI: Body mass index, CAD: Coronary artery disease, CVA: Cerebrovascular accident, PAD: Peripheral arterial disease,  CKD: Chronic kidney disease, COPD: Chronic obstructive pulmonary disease, AF: Atrial fibrillation, SBP:  Systolic blood pressure,  DBP: Diastolic blood presssure, ACS: Acute coronary syndrome, UA: Unstable angına,  NSTEMI:  Non-ST elevation mycardial infarction, STEMI: ST elevation mycardial infarction.

Table 2. Comparison of patients included and excluded in terms of laboratory and angiographic parameters at the admission

Laboratory analysis

Patients included (n=629)

Patients excluded (n=123)

p value

WBC count, 109/L

11.17±3.95

11.69±3.80

0.184

Lymphocyte, 109/L

2.35±2.81

2.11±1.13

0.352

Hemoglobin, g/dl

13.36±2.21

13.45±2.25

0.682

CRP

4.96±0.20

3.95±0.36

0.642

Glucose, mg/dl

129 (106-178)

133 (108-193.5)

0.251

GFR, ml/min

89 (68.975-108)

87.5 (68.175-111.25)

0.901

TG, mg/dl

128 (92.5-185.5)

128.5 (92.5-174.5)

0.507

TC, mg/dl

183.99±48.25

46.04±4.20

0.557

LDL-C, mg/dl

118.90±35.85

120.90±34.11

0.572

Total bilirubin, mg/dl

0.50 (0.30-0.70)

0.40 (0.40-0.60)

0.338

ALT, IU/L

21 (15-30)

23 (15.5-32.5)

0.131

AST, IU/L

34 (24-58)

34 (25-59)

0.168

INR

1.14±0.29

1.12±0.22

0.566

APTT

29.3 (26.95-33.70)

29.4 (27.00-36.20)

0.190

Albumin

3.68±0.53

3.68±0.58

0.996

Sodium

137.51±3.41

137.58±2.74

0.834

Potassium

4.40±0.58

4.41±0.55

0.876

Calcium

8.97±0.75

9.04±0.62

0.523

Magnesium

1.95±0.35

1.98±0.35

0.628

LVEF, %

51.07±9.73

51.73±9.35

0.504

Extend of CAD*

One-vessel disese

194 (84.72)

35 (15.28)

0.441

Multivessel disease

374 (82.38)

80 (17.62)

Descriptives: mean ±sd, median (25th–75th percentile), count (percent).Abbreviations: WBC: White blood cell, CRP: C- reactive protein, GFR: Glomerular filtration rate, TG: Triglycerides, TC:Total cholesterol, LDL-C:Low density lipoprotein-cholesterol, ALT:Alanine aminotransferase, AST: Aspartate aminotransferase,  INR: International normalized ratio, APTT: Activated partial thromboplastin time, FT3: Free T3 (triiodothyronine),   FT4; Free T4 ( thyroxine), TSH: thyroid stimulating hormone,  LVEF: Left ventricular ejection fraction, CAD: Coronary artery disease, *The number of coronary angiography applied subjects.

Table 3. Comparison of patients included and excluded in terms of drug treatment given in the hospital.

In-hospital medication

Patients included (n=629)

Patients excluded (n=123)

p value

ASA

613 (97.46)

120 (97.56)

0.946

Clopidogrel

482 (76.63)

90 (73.17)

0.411

Ticagrelor

111 (17.65)

27 (21.95)

0.259

Prasugrel

12 (1.91)

2 (1.63)

0.833

Beta-blockers

474 (75.36)

100 (81.30)

0.156

CCB(dihydropyridine)

40 (6.36)

6 (4.88)

0.531

ACEI

380 (60.51)

83 (68.03)

0.118

ARB

6 (0.95)

1 (0.81)

0.882

Heparin (all class)

574 (91.26)

114 (92.68)

0.604

Statins

549 (87.28)

109 (88.62)

0.682

Furosemid

56 (8.90)

5 (4.07)

0.072

Spironolactone

33 (5.25)

7 (5.69)

0.841

Nitrates

125 (19.87)

24 (19.51)

0.927

İnotropic agents

53 (8.43)

12 (9.76)

0.631

Descriptives: count (percent). Abbreviations: ASA: Acetylsalicylic acid, CCB: Calcium channel blocker, ACEI: Angiotensin converting enzyme inhibitör,  ARB: Angiotensin receptor blocker.

Table 4. Comparison of patients included and excluded in terms of advers event development in-hospital and follow-up period.

In-hospital period

Patients included (n=629)

Patients excluded (n=123)

p value

CICU LOS, days

3 (2-4)

3 (2-4)

0.098

Total LOS, days

6 (4-8)

5 (4-6)

<0.001*< span="">

In-hospital event

161 (25.60)

30 (24.39)

0.779

Major bleeding

25 (3.97)

2 (1.69)

0.200

Inhospital transfusion

38 (6.04)

7 (5.69)

0.881

Re-infarct

2 (0.32)

0 (0.00)

0.531

VT/VF

17 (2.70)

3 (2.44)

0.868

Heart failure

15 (2.38)

2 (1.63)

0.605

New onset AF

23 (3.66)

3 (2.44)

0.499

Acute renal failure

30 (4.77)

6 (4.88)

0.959

CPA

48 (7.64)

11 (8.94)

0.624

Mechanic complications

9 (1.43)

2 (1.63)

0.869

Infection

55 (8.74)

12 (9.76)

0.719

In-hospital mortality

55 (6.52)

12 (5.69)

0.731

Inotrophic drug requirement

48 (7.63)

12 (9.76)

0.426

Stroke

2 (0.32)

0 (00.00)

0.531

Total  event

107 (17.01)

20 (16.26)

0.839

Mortality

45 (7.15)

12 (9.76)

0.319

Re-hospitalisation

42 (6.68)

8 (6.50)

0.944

Re-infarct

66 (10.49)

16 (13.01)

0.413

Cumulative mortality

86 (13.67)

19 (15.45)

0.604

Descriptives: median (25th–75th percentile), count (percent). Abbreviations: CICU: Coronary intensive care unit,  LOS: Lenght of stay, VT: Ventricular tachcardia,  VF: Ventricular fibrillation,  AF: Atrial fibrillation,  CPA: Cardiopulmonary arrest.

Round  2

Reviewer 2 Report

Authors have addressed all my comments effectively.